# Crosstalk Between Allergic Inflammation and Autophagy

**DOI:** 10.3390/ijms26199765

**Published:** 2025-10-07

**Authors:** Jaewhoon Jeoung, Wonho Kim, Dooil Jeoung

**Affiliations:** Department of Biochemistry, Kangwon National University, Chuncheon 24341, Republic of Korea; heyjhw@kangwon.ac.kr (J.J.); kimwonho99@kangwon.ac.kr (W.K.)

**Keywords:** allergy, autophagy, crosstalk, mitochondria, mitophagy

## Abstract

Autophagy is a conserved process that involves the degradation of damaged proteins and organelles to restore cellular homeostasis. Autophagy plays a critical role in cell differentiation, immune responses, and protection against pathogens, as well as the development and progression of allergic inflammation. Crosstalk between autophagy and signaling pathways modulates immune responses to inflammatory signals. Here, we discuss the regulatory roles of autophagy in allergic inflammation. Autophagy can promote allergic inflammation by enhancing the secretion of inflammatory mediators. Impaired autophagy resulting from the accumulation of autophagosomes can exacerbate allergic inflammation. Mast cell degranulation and activation require energy provided by mitochondrial respiration. Mast cell activation is accompanied by morphological changes and mitochondrial fragmentation. Mitochondrial fragmentation (mitophagy) induced by oxidative stress involves the degradation of defective mitochondria. Therefore, we discuss the relationship between mitophagy and allergic inflammation. Targeting autophagy and oxidative stress can be a strategy for developing anti-allergy therapeutics. In this review, we also discuss future research directions to better understand allergic diseases with respect to autophagy and develop effective anti-allergy drugs.

## 1. Introduction

Autophagy involves the lysosomal degradation of damaged proteins and organelles and the phagocytosis of foreign bodies (viruses, bacteria, and fungi) in the body. It is responsible for maintaining homeostasis and the survival of normal cells. Thus, autophagy dysregulation leads to various diseases, including allergic diseases, cancers and neurodegenerative diseases [1,2]. Although autophagy is viewed as a degradation pathway, its role in the secretion of allergic inflammation mediators during allergic inflammation has been reported [2]. Allergic rhinitis involves autophagosome formation [3]. Mitochondrial ATP production is required for mast cell activation [4]. Therefore, mitochondrial autophagy (mitophagy) may affect the progression of allergic inflammation. Impaired autophagy and the accumulation of autophagosomes can contribute to the pathogenesis of allergic airway diseases [5]. Thus, autophagy may play a regulatory role in allergic inflammation.

In this review, we aim to describe the mechanisms of autophagy-promoted allergic inflammation, the negative regulatory role of autophagy in allergic inflammation, and the role of mitophagy in allergic inflammation. We also discuss future research directions for developing anti-allergy therapeutics based on a thorough understanding of the mechanisms of allergic inflammation in relation to autophagy.

## 2. Autophagic Processes

Autophagy can be selective and non-selective. Non-selective autophagy does not require specific receptors and involves the degradation of cytoplasmic materials [6]. Selective autophagy, an evolutionarily conserved process, involves the ubiquitination and subsequent degradation of protein aggregates and dysfunctional organelles, along with the generation of specific nutrients in response to environmental changes [7]. Selective autophagy includes the clearance of intracellular pathogens (xenophagy) [8], the degradation of damaged mitochondria (mitophagy), endoplasmic reticulum (ER phagy) [9], peroxisomes (pexophagy) [10], polyubiquitinated aggregates (aggrephagy) [11], ribosomes (ribophagy) [12], lipid droplets (lipophagy) [13], and cilia components (ciliophagy) [14]. Autophagy is classified into macroautophagy, microautophagy, and chaperone-mediated autophagy (CMA). Macroautophagy involves the formation of autophagosomes. Microautophagy involves the degradation of cytoplasmic components by directly engulfing them into lysosomes [15]. CMA employs chaperone proteins to deliver specific proteins to lysosomes [16]. Figure 1 shows the types and classes of autophagy. This review focuses on macroautophagy (autophagy).

P62, a specific receptor of autophagy, regulates autophagy and neuroinflammation [17,18]. Decreases in P62 protein levels are closely associated with autophagy activation [19,20,21]. P62 binds to and facilitates the elimination of protein aggregates. Excess p62 inhibits the clearance of ubiquitinated proteins destined for proteasomal degradation [22]. Autophagy inhibition leads to an increase in the size and number of p62 bodies and p62 protein levels [22,23].

Unc-51-like autophagy-activating kinase 1 (ULK1) is necessary for the initiation of autophagy [24]. The ULK1 kinase complex consists of autophagy-related gene 13 (ATG13), ATG101, and FAK-family interacting protein of 200 kDa (FIP200). AMP-dependent kinase (AMPK) activates ULK1 to initiate phagophore formation [25]. Mammalian target of rapamycin C1 (mTORC1) inhibits ULK1 activity [26,27] (Figure 2). Proteins and lipids are recruited to phagophore assembly sites (PAS). The proteins recruited to PAS include the phosphatidyl inositol 3-kinase (PI3K) complex, which produces phosphatidyl inositol 3-phosphate (PI3P) for phagophore formation. The PI3K complex, also known as PIK3C3-C1, includes type III phosphatidylinositol kinase (VPS34), B-cell lymphoma-2 protein interaction center coiled-coil protein 1 (Beclin 1), ATG14, autophagy, and Beclin 1 regulator 1 (AMBRA1), and p115 [28]. AMBRA1, phosphorylated by ULK1, binds to Beclin 1, and subsequently activates class III phosphatidylinositol 3-kinase complex I (PI3KC3-C1, a catalytic subunit of PI3K) [29] (Figure 2). Activated PI3KC3-C1 is then recruited to PAS by ATG14. Autophagy elongation leads to the expansion of the autophagosome (a double-membrane vesicle) to engulf cellular components. ATG12-ATG5 and microtubule-associated protein 3 (LC3)-phosphatidyl ethanolamine (LC3-PE) mediate autophagy elongation. ATG12 is conjugated to ATG5 via ATG7 and ATG10. ATG7 acts as a ubiquitin-activating enzyme, and ATG10 acts as a ubiquitin-conjugating enzyme. ATG12-ATG5 is then conjugated to ATG16L (Figure 2). ATG4 cleaves newly synthesized LC3 to produce LC3-I. Cleavage by ATG4 exposes C-terminal glycine, which can be covalently attached to phosphatidyl ethanolamine (PE). LC3-I binds to PE to produce LC3-II (Figure 2). ATG12-ATG5-ATG16L acts as a ubiquitin ligase and converts LC3-I to LC3-II along with ATG7 and ATG3 [30]. P62, a specific receptor of autophagy, binds to ubiquitinated proteins and LC3 to deliver cargo to the lysosome. The elongating membrane closes around the cargo and forms an autophagosome. The autophagosome then fuses with the lysosome. The fusion of autophagosomes with lysosomes is mediated by Ras-associated binding proteins (RABs) and the soluble N-ethylmaleimide-sensitive factor attachment protein receptor SNARE complex (Figure 2). Autophagy contributes to the identification and clearance of infectious pathogens [31] and the occurrence and development of cancer, cardiovascular diseases, and allergic diseases [18,32,33]. Autophagy is involved in various cellular processes, such as development, differentiation, and immune responses. Autophagy dysregulation is known to cause various diseases.

## 3. Autophagy and Allergic Asthma

Autophagy plays critical roles in immune cell differentiation [34], antigen presentation [35], and the development of protective immunity against pathogens [36]. CD8^+^ T cells require autophagy to recycle internal proteins for survival [34]. The blocking of CD8 prior to OVA sensitization can attenuate bronchial hyperresponsiveness [37]. CD8^+^ T cells that express granzyme K promote allergic airway inflammation by activating the complement cascade [38]. Autophagy regulates the expression of major histocompatibility complex II (MHCII) and class II invariant peptide (CLIP) [35]. Autophagy can induce the activation of immune cells such as mast cells, T cells, and dendritic cells (DCs) to promote the occurrence of food allergies [39,40]. 

Bioinformatics analysis shows that patients with severe asthma display high levels of tumor protein 53 (TP53), SQSTM1/p62, and ATG5 compared to those with non-severe asthma and healthy controls [18]. ATG5 polymorphism is closely related to the severity of bronchial asthma [41]. Thus, autophagy-related genes can be used as diagnostic and prognostic markers for allergic diseases.

ATG5 mediates allergic airway inflammation and remodeling during allergic asthma [42]. ATG-deficient mast cells show severely impaired mast cell degranulation in a mouse model of passive cutaneous anaphylaxis (PCA) [43]. Budesonide and simvastatin inhibit mTOR-mediated autophagy in macrophages and asthma by decreasing the expression of Beclin 1 and LC3 [44].

Inhibiting autophagy reduces eosinophilia in asthma models [45]. The most prevalent form of asthma is classified as T2-high asthma. T2-high asthma is driven by eosinophils, mast cells, and T-helper 2 (Th2) cells. Th2 cells contribute to the pathogenesis of eosinophilic asthma (EA), whereas Th1 and Th17 cells play a dominant role in neutrophilic asthma (NA). Th2 cytokines, such as interleukin-4 (IL-4), IL-5, and IL-13 mediate allergic T2-high asthma [46]. The level of autophagy is higher in Th2 cells than in Th1 cells [47]. These reports indicates that Th2 cytokines can mediate allergic asthma by promoting autophagy.

Mice with NA exposed to endocrine-disrupting mono-n-butyl phthalate (MnBP) display increased airway hyperresponsiveness, more neutrophils in bronchoalveolar lavage (BAL) fluid, and a higher percentage of M1 macrophages in lung tissue compared to mice not exposed to MnBP [48]. Hydroxychloroquine, an autophagy inhibitor, reduces the effects of MnBP in vivo and in vitro [48]. MnBP exposure may increase the risk of neutrophilic inflammation in severe asthma, and autophagy pathway-targeted therapeutics can help control MnBP-induced harmful effects in NA. Taken together, these results suggest that autophagy contributes to the pathogenesis of allergic asthma.

## 4. Mechanisms of Autophagy-Mediated Allergic Inflammation

Allergens can trigger autophagy in immune cells such as mast cells [49,50], potentially impacting the severity and progression of allergic responses. 

The nasal mucosa of mice with allergic rhinitis induced by OVA-loaded BMDCs displays increased expressions of LC3-II and ATG5 and decreased p62 expression [39]. The activation of AMPK and the inhibition of mTOR contribute to the pathogenesis of allergic rhinitis [51]. Allergic rhinitis involves increased levels of autophagy-related proteins in innate lymphoid cells (ILC2) [52] and autophagy in DCs [53]. LC3-II is co-expressed with CD11c^+^ DC markers. The inhibition of autophagy in DCs decreases the levels of IgE, leukotriene C4 (LTC4), eosinophil cationic protein (ECP), and prostaglandin D2 (PGD2) in a mouse model of allergic rhinitis [54]. Brain-derived neurotrophic factor (BDNF) promotes the proliferation of human airway smooth muscle cells during asthma progression by inducing autophagy [55]. The expression of Wnt5a is increased in asthma patients compared to healthy controls [56]. Wnt5a promotes epithelial–mesenchymal transition (EMT) and autophagic flux in human bronchial epithelial cells (HBECs) during asthma progression [56]. It will be necessary to further employ clinical samples or patient data to understand the role of autophagy in allergic inflammation.

OVA and papain induce allergic asthma by increasing the expression of ATG5 and LC3-II via the transcription factor EB (TFEB) [57]. Airway remodeling in allergic asthma is accompanied by the increased phosphorylation of AMPK and the expression of silent information regulator 1 (SIRT1), a NAD^+^-dependent protein deacetylase and peroxisome proliferator-activated receptor gamma coactivator 1α (PGC-1α) [58]. SIRT1-induced deacetylation of FOXO3, Beclin 1, ATGs and LC3 leads to autophagy induction [59]. 2,4-Dinitochlorobenze (DNCB)-induced mast cell activation increases the expression of SIRT1 [60]. Forkhead box O3a (FOXO3a) polymorphism is closely associated with asthma severity [61]. FOXO3a enhances intracellular autophagic flux in thrombin-treated HT-22 hippocampal neuronal cells [62]. Mast cell activation contributes to the pathogenesis of various inflammatory diseases [63,64]. Decreased mTORC1 activity causes the translocation of TFEB to the nucleus, which, in turn, enhances mast cell degranulation [65]. This suggests that autophagy promotes mast cell activation. Since autophagy is responsible for mast cell activation, unstimulated mast cells might protect against allergic inflammation. Unstimulated mast cells show protective effects against OVA-induced asthma in rats by decreasing the expression of LC-3B and Beclin 1 [66].

Allergic asthma increases the expression of wingless-type MMTV integration site (Wnt) [66,67]. Wnt3a increases the expression of Th2 cytokines and enhances histamine and tryptase release in human mast cells [67]. Wnt 5a promotes autophagy and EMT in HBECs during asthma progression [56]. Wnt signaling promotes mast cell maturation by increasing the expression of histone decarboxylase [68]. Wnt5a mediates LPS-induced inflammatory disease by regulating the expression of Beclin1, LC3-II, and p62 [69]. IL-4 enhances the permeability of vascular endothelial cells by activating Wnt5a signaling [70]. Wnt signaling can enhance the response to IL-4 in macrophages [71]. This further confirms the role of Wnt signaling in allergic inflammation. Thus, Wnt signaling can mediate allergic inflammation by promoting autophagy.

The EMT of HBECs is critical for airway remodeling during asthma [72]. Integrin β3 (ITGB3) protects against asthma by inhibiting EMT through suppression of the NF-κB signaling pathway [72]. The mast cell-specific deletion of E-cadherin induces anaphylaxis [73]. Chymase, secreted by activated mast cells, mediates airway remodeling by decreasing the expression of E-cadherin and matrix metalloproteinases [74]. Thus, allergic airway inflammation is mediated by extracellular matrix remodeling and EMT. These reports confirm the role of EMT in allergic inflammation and autophagy.

Toll-like receptor 2 (TLR2) induces autophagy through PI3K/AKT to promote airway inflammation, and luteolin attenuates allergic asthma by inhibiting autophagy via PI3K/AKT/ mTOR pathway activation [75]. TLR-2 is necessary for the activation of DCs by IL-4 secreted by mast cells [76]. Recombinant profilin induces allergic airway responses by increasing the expressions of TLR2 and Th2 cytokines [77].

The lack of autophagy in type 2 innate lymphoid cells (ILC2) promotes glycolysis, inhibits fatty acid oxidation and the tricarboxylic acid (TCA) cycle, suppresses Th2 cytokine production, and attenuates asthma airway hyperresponsiveness [78]. In house dust mite (HDM)-induced allergic asthma, leukotriene B4 receptor (BLT1) is responsible for the increased expression of Th2 cytokines. M2 polarization of macrophages was predominant and the numbers of Th2 cells and eosinophils increased [79]. BLT1 antagonist U75302 inhibits autophagy and chronic inflammation in cigarette smoke-induced chronic obstructive pulmonary disease (COPD) [80]. Thus, BLT1 can regulate allergic airway inflammation by increasing autophagy and the production of Th2 cytokines.

OVA stimulation increases the number of autophagosomes, the expression of autophagy-related proteins in BMDCs and the development of allergic rhinitis in C57BL/6 mice [39]. The nasal mucosa of mice in the allergic rhinitis group shows increased LC3-II and ATG5 expression and decreased p62 expression [39]. LC3-II is co-expressed with CD11c^+^ DC markers [39]. Thus, allergens can induce autophagy in CD11c^+^ DCs and affect the immune imbalance of downstream T cells toward allergic inflammation.

Thymic stromal lymphopoietin (TSLP) contributes to the pathogenesis of allergic inflammation [81,82,83]. Tezepelumab, a fully human monoclonal IgG2 antibody against TSLP, decreases IL-5, IL-13, and IgE levels in severe asthma [84]. TSLP suppresses sepsis-induced liver injury by inducing autophagy via PI3K/Akt/STAT3 signaling [85]. TSLP increases the levels of inflammatory cytokines by inducing autophagy [86]. TSLP increases the expression levels of Beclin 1, LC3-II, p62, and ATG5, via JAK1/JAK2/STAT5/JNK/PI3K signaling [86]. CQ, an inhibitor of autophagy, decreases ear thickness, skin damage, mast cell numbers, and the levels of IgE, TSLP, IL-4, and IL-13 in a mouse model of atopic dermatitis (AD) [87]. These reports suggest that targeting TSLP can ameliorate allergic diseases by inhibiting autophagy. Targeting TSLP has shown clinical benefits in allergic conditions, including asthma, atopic dermatitis, and food allergies [88]. Figure 3 shows the FcεRI signaling pathways associated with autophagy activation.

## 5. Autophagy Inhibits Allergic Inflammation

Patients in the low-autophagy subtype group display more severe, glucocorticoid-resistant, and poorly controlled asthma [89]. This implies that autophagy can protect against asthma. Complete autophagy may lead to cell death or exacerbate allergic inflammation. Basophil-mediated chronic allergic inflammation involves the formation of protein aggregates [90]. Autophagy involves the degradation of protein aggregates [91]. Thus, autophagy might protect against allergic inflammation by promoting the degradation of protein aggregates.

Src homology protein 2 (SHP2) (SHP2)/JNK signaling pathways mediate airway remodeling in a mouse model of HDM-induced asthma [92]. Severe asthma is accompanied by decreased neutrophil autophagy [93]. The downregulation of SHP2 attenuates asthma by promoting autophagy via the extracellular regulated kinase 5 (ERK5) pathway [93]. The downregulation of SHP2 expression in neutrophils increases the expression of Beclin1 and LC3, whereas the expression levels of p62, cit-H3, myeloperoxidase (MPO), elastase, neutrophil expressed (ELANE), protein-arginine deiminase type-4 (PADI4), and ERK5 are decreased [93]. Thus, neutrophil autophagy can protect against asthma.

Cannabinoids causes tolerogenicity in human DCs by inhibiting mTOR signaling pathways while inducing the activation of AMPK and functional autophagy flux via cannabinoid receptor 1 (CB1) and peroxisome proliferator-activated receptor alpha (PPARα) [94]. Forkhead box P3 (FOXP3)^+^ T-regulatory cells (Tregs) suppress anaphylactic reactions in a mouse model of peanut allergy by inducing tolerogenicity in DCs [94]. AMPK activators prevent antigens from increasing the phosphorylation of ERK, Jun N-terminal kinase (JNK), I kappa kinase (IKK), mTOR, and ribosomal S6 kinase (S6K) [95]. The activation of AMPK negatively regulates FcεRI-dependent mast cell activation in a mouse model of anaphylaxis [95,96,97]. These findings suggest that autophagy can inhibit allergic inflammation. mTOR regulates the pathogenesis of allergic diseases by inhibiting autophagy [27,98,99].

Allergic reactions are accompanied by the increased expression of Th2 cytokines [49,100]. IL-27 attenuates allergic asthma by decreasing the phosphorylation of pAKT, which, in turn, leads to enhanced autophagy [101]. IL-27 attenuates allergic asthma by decreasing the expression of Th2 cytokines [102].

IL-33, elevated in a model of allergic rhinitis, inhibits autophagy and promotes mast cell degranulation via ST2 (a member of the interleukin-1 receptor family)/PI3K/mTOR signaling pathway [103]. Azithromycin inhibits the proliferation of airway smooth muscle cells by inducing autophagy [104]. Cellular interactions contribute to the pathogenesis of allergic inflammation [49]. It is probable that IL-33 is present in exosomes to promote allergic inflammation by inducing cellular interactions. Diesel exhaust (DE)-promoted lung epithelial cell inflammation is accompanied by the increased production of IL-33 and oxidative stress [105]. This suggests that reactive oxygen species (ROS) can promote autophagy during allergic inflammation induced by IL-33. ROS promotes autophagy by increasing the expression of ATG4 [106].

Resveratrol (RSV) regulates the expression of LC3, Beclin1, and p62 to promote autophagy and attenuate renal inflammation by upregulating SIRT1 expression in Wistar rats [107]. This suggests that SIRT1 might inhibit allergic inflammation. Suzi Daotan Decoction (SZDTD) attenuates allergic asthma by increasing the expression of pAMPK and SIRT1 while decreasing the expression of pAKT and Th2 cytokines [58]. Palmatine (PAL), the main chemical component of Yajieshaba, alleviates skin lesions associated with food allergies by increasing the expression of LC3, Beclin 1, p-AMPK, and ATG-5 [95].

Treatment with the proteosome inhibitor MG132 attenuates 2,4-dinitrofluprobenzene (DNFB)-induced AD by decreasing the number of Th17 cells, serum IgE production, and mast cell migration [108]. The proteosome inhibitor bortezomib abolishes hazelnut allergen-specific IgE, accompanied by a decrease in both CD19(+) and follicular B lymphocytes in a mouse model of food allergy (anaphylaxis) [109]. Proteasome inhibitors can increase the expression of tumor necrosis factor receptor associated factor 6 (TRAF6) [110]. TRAF6 binds to p62 and mediates selective autophagy by inducing the degradation of antioxidative enzyme glutathione peroxidase 4 (GPX4) [111]. Thus, proteasome inhibitors can inhibit mast cell activation by inducing autophagy.

Clemastine, a histamine H1R antagonist, inhibits mast cell activation in myocardial ischemia–reperfusion injury [112]. Clemastine attenuates subarachnoid hemorrhage by increasing the expression of autophagy-related genes via nuclear translocation of nuclear factor erythroid 2-related factor 2 (Nrf2) [113]. Thus, allergic inflammation might be accompanied by decreased autophagy. Akebia saponin D increases the expression of pAMPK in human lung epithelial BEAS-2B cells [114] and inhibits the infiltration of various leukocytes, and the production of Th2 cytokines in a mouse model of asthma [114]. These reports suggest that allergic inflammation can be negatively regulated by autophagy.

Allergic inflammation can inhibit AMPK, which, in turn, leads to the recruitment of members of the RAS oncogene family (RAB27 and RAB37) to secretory granules to aid in the secretion of inflammatory mediators during allergic inflammation (Figure 4). AMPK prevents LC3-II from recruiting tripartite motif-containing protein 16 (TRIM16) and ER-localized vesicle-associated SNARE protein SEC22B (Figure 4). Allergic inflammation induces the translocation of mitochondrial microphthalmia-associated transcription factors (MITF) and TFEB, which, in turn, increases the expression of genes for further secretory granule synthesis (Figure 4). Thus, allergic inflammation can inhibit complete autophagy to avoid cell death. Figure 4 illustrates the inhibitory role of autophagy in allergic inflammation.

## 6. Impaired Autophagy in Allergic Inflammation

Both impaired and complete autophagic processes can contribute to the pathogenesis of various human diseases [115]. Impaired autophagy results in the accumulation of autophagosomes and fails to clear dysfunctional cellular components, which enhances susceptibility to infections [115]. Impaired autophagy involves the increased expression of p62, which leads to the accumulation of autophagosomes. Complete autophagy involves the fusion of an autophagosome with a lysosome, which then degrades the intracellular contents [19,20,21]. Allergic rhinitis is characterized by impaired autophagy in lung and olfactory epithelium [53]. Paeoniflorin (PF) alleviates asthma by inhibiting complete autophagy in airway epithelial cells [116]. In doing so, PF activates mTORC1 by promoting raptor-mTOR interactions in airway epithelial cells [116].

Antigen expression increases the expression of pBeclin1, LC3-II, p62, pAMPK, and ATG-5 in RBL2H3 cells [49]. The increased expression of p62 might result from the failure of autophagosome-lysosome fusion and the accumulation of autophagosomes. Since antigen stimulation increases p62 expression, autophagy might mediate allergic inflammation. P62, increased by antigen stimulation, mediates allergic inflammation in vitro, in PCA, and in passive systemic anaphylaxis (PSA) [49]. P62 colocalizes with LC3-II on the exosomes of RBL-2H3 cells [49]. This suggests the role of p62 in mediating cellular interactions during allergic inflammation.

Chronic asthma is associated with an increased expression of p62 and insufficient autophagic flux in a mouse model [117]. The downregulation of p62 decreases the proliferation and migration of human bronchial smooth muscle cells and lactate production in airway smooth muscle cells [117]. Severe food allergies involve the increased production of lactate [118]. Since autophagy decreases the production of lactate, food allergies might involve impaired autophagy.

The effect of p62 on AD-like skin lesions induced by the deletion of JunB/AP-1 was determined by inactivating p62 in JunB^Δep^p62^−/−^ double-knockout mice [119]. The elevated expression of p62 is seen in skin lesions of JunB^Δep^ mice, resembling the upregulation of p62 in AD and psoriasis [119]. Thus, AD-like skin lesions are mediated by impaired autophagy. When p62 is inactivated, JunB^Δep^-associated defects, such as epidermal thickening, skin infiltration by mast cells and neutrophils, and the development of macroscopic skin lesions, are reduced [119]. P62 is necessary for activating mTOR and NF-kB in JunB^Δep^ but not in JunB^Δep^p62^−/−^ double-knockout skin, suggesting an important role of impaired autophagy in AD-like inflammation [119]. P62-dependent signaling pathways may be promising therapeutic targets to attenuate the skin manifestations of AD and allergic diseases that are mediated by impaired autophagy.

## 7. Mitophagy and Allergic Inflammation

The mitochondrial membrane potential generated by the electron transport chain (ETC) can exert effects on mast cell degranulation by providing ATP [120]. Mast cell activation can provide metabolites for glycolysis and the TCA cycle [121]. AD increases the production of free fatty acids such as linoleic and linolenic acid [122]. The excessive uptake of oleic acid exacerbates OVA-induced allergic airway inflammation and induces the M2 polarization of macrophages [123]. Acetyl CoA, a product of fatty acid oxidation, induces autophagy by increasing the expression of autophagy-related genes [124]. Pyruvate dehydrogenase (PDH), a key regulator of the TCA cycle, connects glycolysis to TCA and is necessary for ATP production and mast cell degranulation [125]. MITF binds to PDH and mediates mast cell activation [125]. Inhibiting ERK1/2 suppresses the binding of MITF to PDH, which leads to decreased mast cell activity [126]. These findings suggest the role of mitochondria in allergic inflammation and autophagy.

Antigen stimulation increases the phosphorylation of MITF via ERK1/2 in RBL2H3 cells and BMMCs [126]. The inhibition of ERK1/2 leads to decreased mast cell activity [100,126]. Oxidative damage and defects in mitochondrial respiration contribute to the pathogenesis of anaphylaxis [127,128,129,130]. Phosphorylation of the two transcriptional factors STAT3 and MITF lead to the production of ATP and mitochondrial ROS and promotes mast cell degranulation (Figure 5) [127]. Thus, mitochondrial ATP production can contribute to the pathogenesis of allergic inflammation.

IgE and non-IgE mast cell stimulation can enhance oxidative phosphorylation (OXPHOS) through complexes I and III, leading to increased mitochondrial ATP production and mast cell degranulation [126,131]. Rotenone, an inhibitor of complex I, inhibits mast cell degranulation and decreases the secretion of IL-6, IL-13, TNF-α, and GM-CSF in skin mast cells [122]. Mitochondrial respiratory complex chain inhibition reduces IgE- and non-IgE-mediated mast cell degranulation [132,133]. The uncoupler carbonyl cyanide-4-(trifluoromethoxy)phenylhydrazone (FCCP) inhibits mast cell degranulation by blocking the generation of mitochondrial membrane potential [134]. The inhibition of uncoupling protein 2 (UCP2) enhances histamine release and vascular permeability in mice after allergic and non-allergic stimulation [135]. The downregulation of UCP2 increases the production of mitochondrial ROS [136]. Thus, mitochondrial ROS can modulate allergic inflammation in association with autophagy.

Anaphylaxis involves increased ROS production via growth arrest and DNA damage-inducible 45 (Gadd45)/MEKK4/JNK [137]. Antigens increase ROS production through NADPH oxidase (NOX) and mitochondrial respiration [127]. Antigen stimulation through FcεRI promotes the activation of phospholipase Cγ (PLCγ) and PI3K [138]. PI3K promotes ROS production through NOX activity. N-acetylcysteine, an antioxidant, decreases Th2 cytokine levels, such as IL-4, IL-5, and IL-13, and inhibits allergic airway inflammation in a mouse model [139]. NOX activity is necessary for increasing tryptase and β-hexosaminidase levels during mast cell activation [140]. ROS induce mast cell degranulation through Ca2^+^ flux from the endoplasmic reticulum and extracellular influx by the calcium releasing–activating channel (CRAC) [141,142]. These reports suggest a close relationship between ROS and allergic inflammation.

Electrons can leak from the ETC and induce the production of mitochondrial ROS (MtROS) via PKC-β [143]. MtROS-induced histamine release is decreased by a protein kinase C (PKC) inhibitor [144]. Increased ROS levels activate the AMPK-ULK1 axis, which, in turn, induces PTEN-induced kinase 1 (PINK)/Parkin-dependent mitophagy [25]. PINK/Parkin-mediated mitophagy involves the ubiquitination of mitochondrial proteins. The substrates of Parkin (E3 ubiquitin ligase) include mitochondrial Rho GTPase (MIRO), mitofusin 1, translocase of the outer mitochondrial membrane 20 (TOM20), and TOM22. MIRO acts as a direct target of Parkin. Optineurin (OPTN) and nuclear domain 10 (NDP52) act as selective autophagy receptors by binding to LC3 and mediating ubiquitin-dependent mitophagy [145]. NDP52 acts as redox sensor to mediate mitophagy [146].

Mast cell activation induces mitochondrial fragmentation, morphological changes, and the translocation of mitochondria from the perinuclear region to the cell surface [133]. Mitochondria-targeting curcuminoids inhibit mast cell activation by inducing mitochondrial fragmentation [147]. Mitochondrial fragmentation by these curcuminoids is accompanied by AMPK activation [147]. Peanut allergy results from mitochondrial dysfunction, including increased ROS production in liver cells [148]. Mitochondrial defects involve the increased expression of LC-3 II, Parkin, and PINK1, and the decreased expression of p62 [149]. Mitochondrial DNA (mtDNA) released from cells with mitochondrial defects can act as a danger signal, further contributing to allergic inflammation [150]. PKC and AMPK play critical roles in mitochondrial fission, which is the division of mitochondria [151]. Mitochondrial fission leads to mitochondrial fragmentation. Th2 cytokines can activate AMPK and induce the M2 polarization of macrophages and mitophagy [152]. Thus, allergic inflammation might cause mitochondrial defects via Th2 cytokines.

Mitophagy removes dysfunctional mitochondria and is an essential process contributing to mitochondrial quality control. Mitochondrial fission is a prerequisite for mitophagy. Mitochondrial fission can generate fragments that are then targeted for removal by mitophagy. OVA-induced allergic asthma involves mitochondrial dysfunction, such as reduced ATP production, ROS accumulation, and mitophagy activation (increased expression of PINK1) [153].

Mast cell degranulation leads to the activation of Ras homolog family member A (RhoA) and Ras-related C3 botulinum toxin substrate 1 (Rac1) [154,155,156]. The interaction between PKCδ and Rac1 mediates allergic inflammation in vitro [157]. Constitutive active RhoA induces PINK1 accumulation at mitochondria [158]. This is accompanied by the translocation of Parkin to the mitochondria and the ubiquitination of mitochondrial proteins, leading to the recognition of mitochondria by autophagosomes and their lysosomal degradation [158]. Active RhoA localizes at mitochondria and interacts with PINK1 to enhance PINK stability by inhibiting the cleavage of PINK [158]. Thus, RhoA activation induces mitophagy during allergic inflammation.

Stanniocalcin 2 (STC2) mediates mitophagy and further exacerbates asthma induced by fine particulate matter (PM_2.5_) [159]. In doing so, STC2 increases the expression of SQSTM1 (p62) by increasing its stability [154]. Serum levels of STC2 in asthma patients are higher during periods of high PM2.5 [159]. This implies that impaired mitophagy might contribute to the pathogenesis of asthma, consistent with the fact that autophagy can protect against asthma [80].

The inhibition of mitophagy by long non-coding RNA (lncRNA NEAT1) leads to the progression of asthma by enhancing the proliferation of airway smooth muscle cells and promoting the production of inflammatory cytokines in asthma patient-derived airway smooth muscle cells [160]. The inhibition of mitophagy by sodium metabisulfite promotes mast cell degranulation [161]. These reports suggest that mitophagy can also inhibit allergic inflammation. Figure 5 shows the mechanisms of allergic inflammation-promoted mitophagy.

## 8. Discussion and Perspectives

P62, increased during impaired autophagy, can mediate mitophagy by binding to LC3 [162]. MicroRNAs play critical roles in various allergic diseases [163]. miR-135 binds to the 3′UTR of p62 to decrease the expression p62 [49]. Thus, small molecules that can bind to and inhibit p62 should be identified, and miR-135 mimic might be developed as an anti-allergy drug.

Allergic inflammation is accompanied by cellular interactions [164]. Histone deacetylase 6 (HDAC6) mediates cellular interactions during allergic inflammation [165]. HDAC6 contains a ubiquitin-binding domain and can target misfolded and damaged organelles for destruction by autophagy [166]. HDAC6 is necessary for autophagosome maturation and lysosomal fusion [166,167]. The phosphorylation of HDAC6 by type III phosphatidylinositol kinase (VPS34) leads to the formation of an autophagosome [168]. Thus, novel HDAC6-specific inhibitors may ameliorate allergic diseases. miRNA mimics and/or miRNA inhibitors that can regulate the expression of HDAC6 can also be developed as anti-allergy drugs.

HDAC6 promotes mitophagy by decreasing the acetylation of tubulin [169]. The deacetylation of p53 by HDAC6 is necessary for maintaining BCL2/adenovirus E1B 19 kDa protein-interacting protein 3 (BNIP3) expression during mitophagy [170]. BNIP3 mediates ubiquitin-independent mitophagy [171]. Since mitochondrial dysregulation plays an important role in allergic inflammation, targeting HDAC6 may modulate allergic inflammation. HDAC6 can bind to and degrade ATG3 [172]. Thus, the effects of HDAC6 on the expressions of autophagy-related genes should be further investigated.

For a better understanding of allergic inflammation, autophagy should be examined in various immune cells, such as DCs, macrophages, eosinophils, basophils, T cells, B cells, and mast cells, during the progression of allergic inflammation to better understand allergic inflammation and develop safe and effective anti-allergy drugs.

The increased expression of autophagy-related genes does not necessarily predict the occurrence of autophagy-promoted allergic inflammation. Therefore, autophagy-specific markers must be identified through unbiased screening to validate the clinical utility of modulating autophagy.

The role of selective autophagy involving various organelles, including the nucleus, ER, and Golgi apparatus must be investigated to better understand the mechanisms of autophagy-regulated allergic inflammation.

Extensive studies on oxidative stress and autophagy are needed to better understand the mechanisms of allergic diseases in association with autophagy. Mast cell activation produces ROS, which, in turn, causes lipid peroxidation to promote allergic asthma [173]. Lipid peroxidation is responsible for ferroptosis, a form of cell death regulated by iron [174]. Dexamethasone promotes ferroptosis via AMPK, ATG5, and ATG [175], suggesting a close relationship between ferroptosis and autophagy. The blocking of IL-33 ameliorates asthma by suppressing ferroptosis [176]. L-malic acid attenuates HDM-induced asthma by inhibiting ferroptosis [177]. OVA-induced allergic asthma involves ferroptosis in bronchial epithelial cells [178]. Mitochondrial ROS induces ferroptosis, which, in turn, leads to autophagy [179]. Thus, studies aimed at understanding the relationship between ferroptosis and allergic inflammation will present reliable targets for developing anti-allergy drugs.

Autophagy, oxidative stress, metabolism, and mitochondrial dysfunction act concurrently or synergistically to promote allergic diseases. Extensive clinical trials of antibodies, chemicals, miRNAs, and SiRNAs targeting autophagy and mitophagy will be helpful for developing effective and safe anti-allergy drugs. Figure 6 describes future research directions for developing anti-allergy drugs.

## Figures and Tables

**Figure 1 ijms-26-09765-f001:**
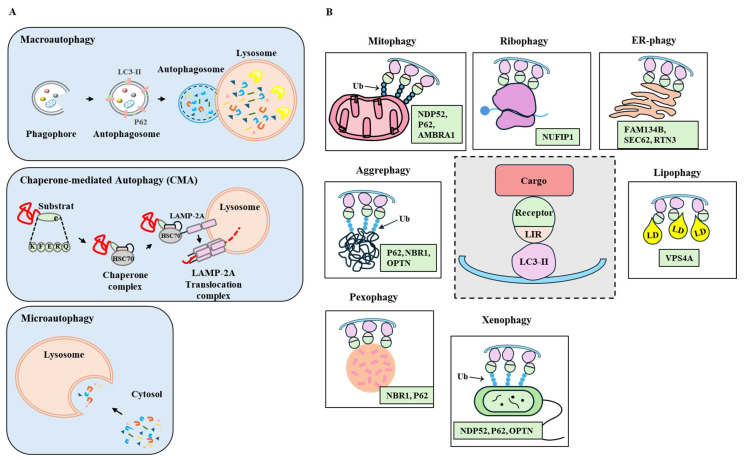
(**A**) Main types of autophagy include macroautophagy, microautophagy, and chaperone-mediated autophagy. Macroautophagy involves the formation of an autophagosome, a double membrane vesicle that engulfs and sequesters to deliver cargo to a lysosome for degradation. In microautophagy, lysosomal membrane is invaginated to uptake cargo. Chaperone-mediated autophagy (CMA) involves the transport of individual unfolded protein into a lysosome with the help of chaperone proteins and lysosomal receptors. Chaperone proteins, such as heat shock cognate protein 70 (HSC70) mediate CMA. Lysosomal-associated membrane protein-2A (LAMP-2A) binds to hsc70 protein and substrate to degrade the substrate. Arrows denote the direction of reaction. (**B**) Selective autophagy involves the formation of an autophagosome, which then fuses with lysosome to degrade intracellular contents. Selective autophagy is different from bulk autophagy in that selective autophagy focuses on particular substrates, such as damaged organelles (mitochondria, ER, peroxisome), protein aggregates (aggrephagy), or pathogens (xenophagy). Unlike bulk (non-selective) autophagy, selective autophagy utilizes selective autophagy receptors. Selective autoaphgy receptors bind to LC3-II via LC3-interatiing regin (LIR). LD denotes lipid droplet. Selective autophagy receptors are shown: FAM134B, Family With Sequence Similarity 134 Member B; NBR, Neighbor of BRCA1 gene 1; NUFIP1, Nuclear FMR (Fragile X Messenger Ribonucleoprotein 1) interacting protein 1; RTN3, Reticulon-3; SEC62, Translocation Protein SEC62; VPS4A, Vacuolar Protein Sorting 4 Homolog A. Ub denotes ubiquitinated proteins.

**Figure 2 ijms-26-09765-f002:**
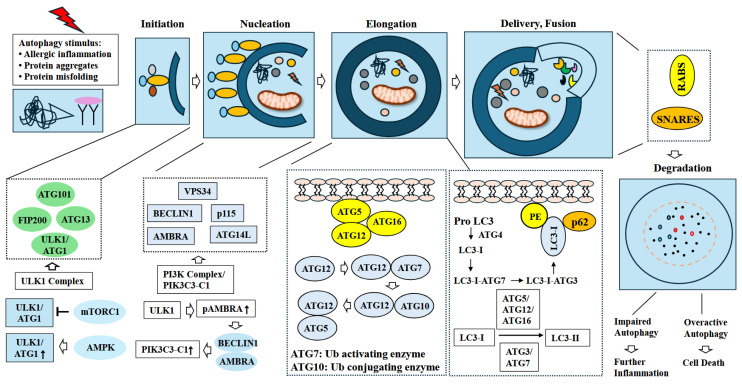
Mechanisms of autophagy activation. Autophagy occurs in response to various stimuli, including allergic inflammation, protein misfolding and protein aggregates. Autophagic process consists of initiation, nucleation, elongation, maturation, delivery, and fusion of autophagosomes and lysosomes. ULK1 is necessary for the initiation. ULK1 is activated by AMPK. MTOR negatively regulates ULK1 activity. ULK1 kinase complex includes ATG13, ATG101, and FIP200. Nucleation involves recruitment of phosphatidyl inositol 3-kinase (PI3K) complex to phagophore assembly site. AMBRA1, phosphorylated by ULK1, binds to and activates Class III phosphatidylinositol 3-kinase complex I (PI3KC3-C1; catalytic subunit of PI3K) by binding to Beclin1. ATG14 recruits activated PI3KC3-C1 to PAS. ATG12-ATG5 and LC3 and LC3-PE mediate autophagy elongation (expansion of autophagosome). The conversion of LC3-I to LC3-II (LC3-PE) is mediated by ATG7 and ATG3. P62 is necessary for efficient delivery of cargo to lysosome. RABs and SNARE complexes mediate fusion of autophagosome with lysosomes. Hollow arrows and arrows denote the direction of reaction. T bar denotes the inhibition of reaction. ↑ denotes increase in activity/expression.

**Figure 3 ijms-26-09765-f003:**
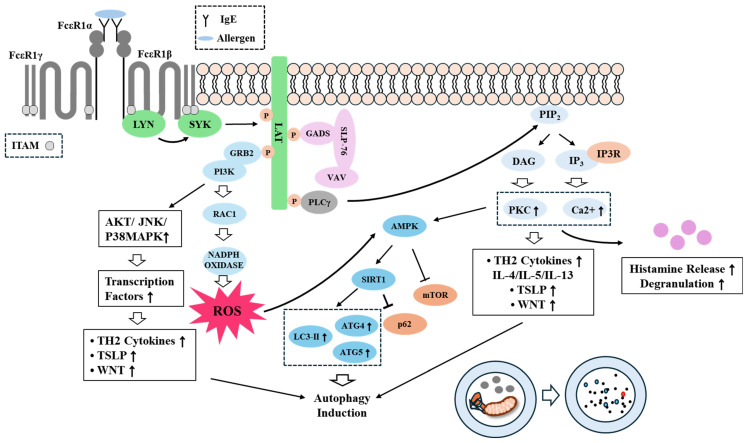
Allergic inflammation promotes autophagy. Allergic inflammation induces the activation of FcεRI signaling. Upon antigen stimulation, LYN phosphorylates β and γ subunits (ITAM) of FcεRI. The phosphorylated ITAM recruits SYK, which in turn phosphorylates linker of activated T cells (LAT). Activation of LAT induces activation of phospholipase Cγ (PLCγ) and PI3K. The activated PLCγ leads to the production of inositol 1,4,5 triphosphates (IP3) and diacyl glycerol (DAG). DAG activates PKC. IP3, through binding to IP3 receptor (IP3R) on ER and releases calcium from stores. Activated PKC and calcium mobilization induce histamine release and increases the production of Th2 cytokines. PI3K activates rac1 to enhance mast cell migration. PI3K increases the production of ROS by activating NADPH oxidase. ROS and PKC activates AMPK and increases the expression of SIRT1, which in turn induce autophagy, by increasing the expressions of ATG4, ATG5, and LC3 while decreasing the expression of p62. AMPK inhibits mTOR signaling. Th2 cytokines can activate wnt signaling, which can enhance autophagy and contribute to allergic inflammation. TSLP can also enhance autophagy and mediate allergic inflammation. Hollow arrows and arrows denote the direction of reaction. T bar denotes the inhibition of reaction. ↑ denotes increase in activity/expression. ROS denote reactive oxygen species. ITAM, Immunoreceptor tyrosine-based activation motif; GRB2, Growth factor receptor bound protein 2; GADS, GRB2-related adaptor protein 2; SLP-76, SH2 Domain-Containing Leukocyte Protein of 76 KDa; VAV, Vav guanine nucleotide exchange factor.

**Figure 4 ijms-26-09765-f004:**
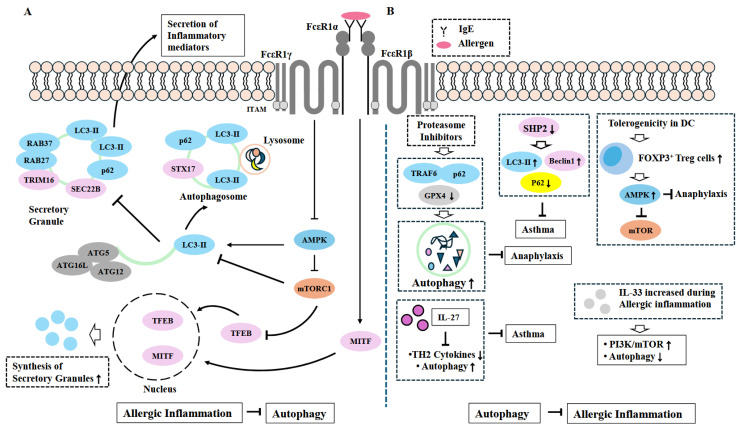
Inhibitory role of autophagy in allergic inflammation. (**A**) Allergic inflammation can inhibit autophagy. Allergic inflammation activates FcεRI signaling. FcεRI signaling inhibits AMPK. The inhibition of AMPK leads to the activation of mTOR signaling. The inhibition of AMPK recruits RAB37, RAB27, TRIM16, and SEC22B to the secretory granule to aid the secretion of inflammatory mediators in mast cells. Arrows denote the direction of reaction. T bars denote the inhibition of reaction. STX17 (Syntaxin-17 protein) plays role for autophagosome fusion with lysosome. T bars denote the inhibition of reaction. Hollow arrows and arrows denote the direction of reaction. (**B**) Autophagy can inhibit allergic inflammation. IL-33 and IL-27, increased during allergic inflammation, can inhibit allergic inflammation by activating autophagy. Proteasome inhibitors increase the expression of TRAF6, which in turn binds to p62 to enhance autophagy. Proteasome inhibitors induce degradation of GPX4. The downregulation of SHP2 inhibits allergic asthma by increasing the expressions of LC3 and Beclin1 while decreasing the expression of p62. Tolerogenicity in DCs inhibits anaphylaxis by promoting autophagy. Induction of tolerogenicity in DCs involves the increased expressions of autophagy-related genes. Clemastine and ASD inhibit mast cell activation by promoting autophagy. Hollow arrows and arrows denote the direction of reaction. T bars denote the inhibition of reaction. ↑ denotes increase in activity/expression. ↓ denotes decrease in activity/expression.

**Figure 5 ijms-26-09765-f005:**
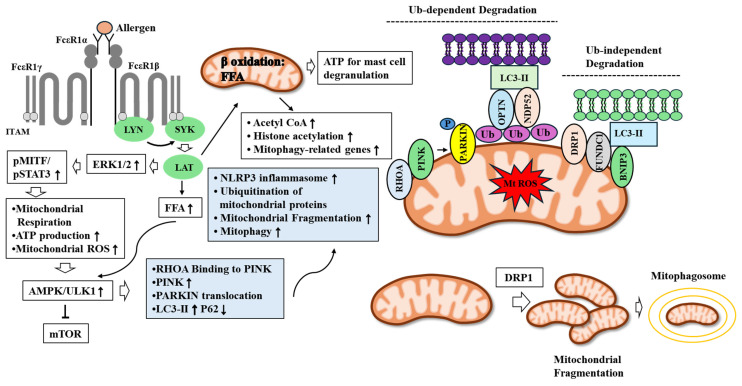
Allergic inflammation-promoted mitophagy. Allergens activate LYN, SYK, and LAT via FcεRI signaling, which activate ERK. Activation of ERK leads to increased phosphorylation of MITF and STAT3. MITF and STAT3 are responsible for mitochondrial respiration, ATP and mitochondrial ROS production. MITF and STAT3 activates AMPK/ULK1 axis. AMPK/ULK1 induces the activation of PINK/Parkin-mediated mitophagy. RhoA binds to PINK to enhance stability of PINK, which in turn translocation of parkin to mitochondria. PKC activates AMPK/ULK1 and PINK/Parkin. AMPK induce inhibitory phosphorylation of Raptor, a key component of mTORC1. Free fatty acids (FFA), produced by mast cell activation, activate PIK3C3/VPS34 kinase complex through AMPK. The β-oxidation of FFAs produce acetyl CoA, acetyl-CoA can induce histone H3 acetylation and increases expressions of autophagy-related genes. PINK/Parkin-mediated mitophagy involves ubiquitination of mitochondrial proteins, mitochondria fragmentation, and NLRP3 inflammasome activation. Ubiquitin-dependent pathway is mediated by mitophagy receptors OPTN and NDP52. Mitophagy also involves ubiquitin-independent pathway that is mediated by mitophagy receptors BNIP3 and FUNDC1. DRP1 induces mitochondrial fission (fragmentation), which leads to mitophagy. Hollow arrows and arrows denote the direction of reaction. T-bar denotes the inhibition of reaction. ↑ denotes increase in activity/expression. ↓ denotes decrease in activity/expression. FFA denotes free fatty acid.

**Figure 6 ijms-26-09765-f006:**
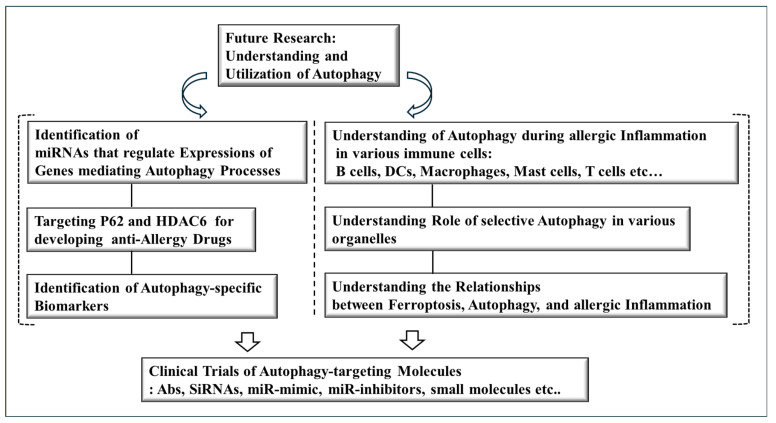
Future research directions for developing anti-allergy drugs. A complete understanding of the relationship between autophagy and allergic inflammation can lead to the successful development of safe and effective anti-allergy drugs. It is also necessary to identify autophagy-specific biomarkers and understand the autophagic processes in various immune cells during allergic inflammation. Hollow arrows denote the direction of reaction.

## Data Availability

No new data were created in this study. Data sharing is not applicable to this article.

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
