# Peer review of "Crosstalk Between Allergic Inflammation and Autophagy"

_ijms, 2025, doi:10.3390/ijms26199765_

Round 1

Reviewer 1 Report

Comments and Suggestions for Authors

The review manuscript by Jeoung et al revises the crosstalk between allergic inflammation and autophagy. This is an interesting topic since autophagy is a key regulatory process that consists on the degradation of proteins and cell components that are redundant, unnecessary or dysfunctional. The manuscript, however, is not easy to read and the text would benefit from the revision of a native English speaker. Here are some suggestions to improve it:

  • Try to condense the technical information in the text describing the molecular events taking place in autophagy and accompany this information with figures, when possible. For example, lines 48-60 describing types and classification of autophagy would benefit from the inclusion of a summarizing figure as well as more references; lines 68-102 could be more concisely described...
  • Include more connectors in the text to avoid repetition of the same terms and give cohesion, e.g. in the abstract, four consecutive sentences start with " Autophagy". This term could be replaced by "this process" or the use of connectors between sentences.
  • Certain parts of the text lack cohesion. Sentences are disconnected and are not coherent. Examples: lines 30-41, lines 116-131, lines 143-157, etc. Ideas and topics are not structured.
  • In line 32 is stated "Autophagy dysregulation leads to various diseases". It would be nice to include examples of such diseases. Are allergic diseases among them? Then, the topic will be introduced.
  • Why do you talk about "allergic inflammations (in plural)? I would rather use the term "allergic diseases/ allergic manifestations" and use "inflammation" only in singular.
  • I think the S1P sections are relevant and you may consider focusing more on this part instead of trying to provide a broad overview of autophagy- allergy connections. 
  • In general, I think the manuscript is relevant and interesting but needs to be rewritten in a more concise and structured way, providing examples and reasons why the information contained is relevant (this is currently lacking), e.g. line 106 "autophagy influences CD8+ T cell differentiation". Why is this relevant in allergic asthma?
  • The discussion and perspectives section, although necessary, should also be shortened and more focused. A table listing the current research needs on this topic may be useful.

Author Response

Dear Sir

Thanks for excellent suggestions. In this revision, I have made necessary changes to accommodate your suggestions. In this revision, I seek professional help for English problems. I include English certificate. In this revision, I remove unnecessary and repetitive sentences to make this manuscript more readable. I added two more figures and many other references to make this manuscript more readable. I also rearrange sentences to make this manuscript more readable. I hope that changes I have made are suitable.      

Sincerely yours

Jeoung Dooil, Ph.D.

Professor of Biochemistry

Kangwon National University

Chuncheon 24341, Korea 

Reviewer 1

Comments and Suggestions for Authors

The review manuscript by Jeoung et al revises the crosstalk between allergic inflammation and autophagy. This is an interesting topic since autophagy is a key regulatory process that consists on the degradation of proteins and cell components that are redundant, unnecessary or dysfunctional. The manuscript, however, is not easy to read and the text would benefit from the revision of a native English speaker. Here are some suggestions to improve it:

Q1. Try to condense the technical information in the text describing the molecular events taking place in autophagy and accompany this information with figures, when possible. For example, lines 48-60 describing types and classification of autophagy would benefit from the inclusion of a summarizing figure as well as more references; lines 68-102 could be more concisely described...

Ans. Thanks. I agree. In this revision, I add new figure (Figure 1). New figure 1 show types and classification of autophagy. Please take look at new figure 1. I hope that this figure displays enough information. I try to make lines 68-102 (new lines 67-98) as concise as possible. However, I would like you to know that I try to be informative as well. I added more references (refs. 8-16) for selective autophagy. I am sorry to cause trouble.

Q2. Include more connectors in the text to avoid repetition of the same terms and give cohesion, e.g. in the abstract, four consecutive sentences start with " Autophagy". This term could be replaced by "this process" or the use of connectors between sentences. In this revision, I let professionals handle English problems.  

Ans. Thanks. I agree. In this revision, I remove unnecessary and repetitive sentences. I hope that this makes this manuscript more readable. In this revision, I shorten abstract to make it more readable. Please take look at new manuscript. In this revision, I avoided four sentences starting with " Autophagy". Please take look at new abstract. I tried to include more connectors throughout this manuscript.  

Q3. Certain parts of the text lack cohesion. Sentences are disconnected and are not coherent. Examples: lines 30-41, lines 116-131, lines 143-157, etc. Ideas and topics are not structured.

Ans. Thanks. I agree. In this revision, I made changes in a way to connect ideas and topics appropriately. For this, I remove unnecessary sentences and revise sentences. Please take look at new manuscript (Lines 29-39; 116-130; 139-152).   

Q4. In line 32 is stated "Autophagy dysregulation leads to various diseases". It would be nice to include examples of such diseases. Are allergic diseases among them? Then, the topic will be introduced.

Ans. Thanks. I agree. I add this sentence (lines 31-33) : Thus, autophagy dysregulation leads to various diseases, including allergic diseases, cancers and neurodegenerative diseases  [1, 2].   

Q5. Why do you talk about "allergic inflammations (in plural)? I would rather use the term "allergic diseases/ allergic manifestations" and use "inflammation" only in singular.

Ans. Thanks. I agree. I changed it as you suggested. Please take look at new manuscript.

Q6. I think the S1P sections are relevant and you may consider focusing more on this part instead of trying to provide a broad overview of autophagy- allergy connections. 

Ans. Thanks. I agree. I understand your concern. In this revision, I made changes to make sure that section 8 describes only allergic inflammation. I am sorry to cause confusion.    

Q7. In general, I think the manuscript is relevant and interesting but needs to be rewritten in a more concise and structured way, providing examples and reasons why the information contained is relevant (this is currently lacking), e.g. line 106 "autophagy influences CD8+ T cell differentiation". Why is this relevant in allergic asthma?

Ans. Thanks. I agree. I add these sentences (lines 101-105): CD8+ T cells require autophagy to recycle internal proteins for survival [34]. The blocking of CD8 prior to OVA sensitization can attenuate bronchial hyperresponsiveness [37]. CD8+ T cells that express granzyme K promote allergic airway inflammation by activating the complement cascade [38]. I hope that this is suitable.     

Q8. The discussion and perspectives section, although necessary, should also be shortened and more focused. A table listing the current research needs on this topic may be useful.

Ans. Thanks. I agree. I added new figure (Figure 7). This figure describes future research directions. In this revision, I tried to make discussion and perspectives section as concise as possible by removing unnecessary and repetitive sentences. Please take look at new section.

Reviewer 2 Report

Comments and Suggestions for Authors

This review addresses an important and timely topic, namely the crosstalk between autophagy and allergic inflammation. The manuscript is generally well-structured and provides a comprehensive overview of recent findings. However, there are several issues that should be addressed before the manuscript can be considered for publication. My comments are as follows:

  1. The distinction between impaired autophagy and overactive autophagy is not clear. For example, on p.8 (L335–344) both terms are used together without clear context, and on p.13 (L572–574) impaired mitophagy is mentioned without linking to earlier notes on excessive autophagy (p.8, L243–244). Please clarify the difference between these two states and explain their roles in different disease settings.
  2. Much of the evidence presented is based on mouse models (e.g., p.5 L152–157; p.6 L237–239; p.13 L570–574). The authors should emphasize the limitations of animal studies and discuss whether findings are supported by clinical samples or patient data.

Author Response

Dear Sir

Thanks for excellent suggestions. In this revision, I have made necessary changes to accommodate your suggestions. In this revision, I seek professional help for English problems. I include English certificate. In this revision, I remove unnecessary and repetitive sentences to make this manuscript more readable. I added two more figures and many other references to make this manuscript more readable. I also rearrange sentences to make this manuscript more readable. I hope that changes I have made are suitable.      

Sincerely yours

Jeoung Dooil, Ph.D.

Professor of Biochemistry

Kangwon National University

Chuncheon 24341, Korea 

Reviewer 2

Comments and Suggestions for Authors

This review addresses an important and timely topic, namely the crosstalk between autophagy and allergic inflammation. The manuscript is generally well-structured and provides a comprehensive overview of recent findings. However, there are several issues that should be addressed before the manuscript can be considered for publication. My comments are as follows:

Q1. The distinction between impaired autophagy and overactive autophagy is not clear. For example, on p.8 (L335–344) both terms are used together without clear context, and on p.13 (L572–574) impaired mitophagy is mentioned without linking to earlier notes on excessive autophagy (p.8, L243–244). Please clarify the difference between these two states and explain their roles in different disease settings.

Ans. Thanks. I agree. I am sorry to cause trouble. In this revision, I remove the terms overactive and excessive autophagy. I would rather use complete (normal) autophagy. Please take look at new manuscript.

I add these sentences (lines 300-305): Impaired autophagy results in the accumulation of autophagosomes and fails to clear dysfunctional cellular components, which enhances susceptibility to infections [115]. Impaired autophagy involves the increased expression of p62, which leads to the accumulation of autophagosomes. Complete autophagy involves the fusion of an autophagosome with a lysosome, which then degrades the intracellular contents [19-21].

  • Complete autophagy means normal autophagy, which involves the decreased expression of p62 and the fusion of autophagosome with lysosome.

Please take look these sentences: Complete autophagy may lead to cell death or exacerbate allergic inflammation (lines 224-225); Thus, allergic inflammation can inhibit complete autophagy to avoid cell death (lines 295-296); Complete autophagy involves the fusion of an autophagosome with a lysosome, which then degrades the intracellular contents [19-21] (lines 303-305); Paeoniflorin (PF) alleviates asthma by inhibiting complete autophagy in airway epithelial cells [117]. In doing so, PF activates mTORC1 by promoting raptor-mTOR interactions in airway epithelial cells [117] (lines 306-308).        

Q2. Much of the evidence presented is based on mouse models (e.g., p.5 L152–157; p.6 L237–239; p.13 L570–574). The authors should emphasize the limitations of animal studies and discuss whether findings are supported by clinical samples or patient data.

Ans. Thanks. I agree. I add these sentences (lines 146-152): Brain-derived neurotrophic factor (BDNF) promotes the proliferation of human airway smooth muscle cells during asthma progression by inducing autophagy [55]. The expression of Wnt5a is increased in asthma patients compared to healthy controls [56]. Wnt5a promotes epithelial-mesenchymal transition (EMT) and autophagic flux in human bronchial epithelial cells (HBECs) during asthma progression [56]. It will be necessary to further employ clinical samples or patient data to understand the role of autophagy in allergic inflammation.

I add this sentence (lines 218-219): Targeting TSLP has shown clinical benefits in allergic conditions, including asthma, atopic dermatitis, and food allergies [88].

I add these sentences (lines 508-511): Serum levels of STC2 in asthma patients are higher during periods of high PM2.5 [198]. This implies that impaired mitophagy might contribute to the pathogenesis of asthma, consistent with the fact that autophagy can protect against asthma [80]. I added some more examples of human data. I hope that these changes are fine.

I added new figure 7 which emphasizes clinical trials.   

Round 2

Reviewer 1 Report

Comments and Suggestions for Authors

The efforts of the authors to improve the quality of the manuscript are appreciated. They have included 2 new figures, modified the text and tried to implement the suggested changes. 

There is however a lack of attention to detail, as it can be seen in the figure caption of new Figure 1, which is full of typos. This denotes authors' impatience and that the final version of the manuscript has not been comprehensively revised. I think this is not acceptable. 

The overall quality of the manuscript has improved as compared with the previous version, but it is, in my opinion, still not satisfactory. I already suggested in my previous revision that the authors may focus on a specific topic:

S1P link between autophagy and allergic inflammation, 

or Autophagy-mediated allergic inflammation vs allergic inflammation inhibited by autophagy

The focus of the review is currently not clear and very broad. Please reconsider a major revision before resubmission.

Author Response

Dear Sir

Thank you for your excellent suggestions. I have made necessary changes to accommodate your suggestions. I hope that changes I have made are suitable. I have learned a lot about autophagy and allergic inflammation from your suggestions. I am thankful for your perspective.

Sincerely yours

Jeoung Dooil, Ph.D.

Professor of Biochemistry

Kangwon National University

Chuncheon 24341, Korea 

Reviewer 1

  1. The efforts of the authors to improve the quality of the manuscript are appreciated. They have included 2 new figures, modified the text and tried to implement the suggested changes. 

There is however a lack of attention to detail, as it can be seen in the figure caption of new Figure 1, which is full of typos. This denotes authors' impatience and that the final version of the manuscript has not been comprehensively revised. I think this is not acceptable. 

The overall quality of the manuscript has improved as compared with the previous version, but it is, in my opinion, still not satisfactory. I already suggested in my previous revision that the authors may focus on a specific topic:

S1P link between autophagy and allergic inflammation, 

or Autophagy-mediated allergic inflammation vs allergic inflammation inhibited by autophagy

The focus of the review is currently not clear and very broad. Please reconsider a major revision before resubmission.

Ans. I thank you for thoroughness. I agree with you. In this revision, I removed sections on S1P-S1PR2. . I also removed references concerning S1P-S1PR2. There is a change in order of references. I removed one figure concerning S1P-S1PR2. In this revision, I wanted to focus on Autophagy-mediated allergic inflammation vs allergic inflammation inhibited by autophagy Please take look at new manuscript. I am sorry to cause confusion. I hope that these changes make this manuscript more readable.

* I checked figure 1 legend. There were so many typos. I took care of these typos. I am sorry to cause trouble.

Reviewer 2 Report

Comments and Suggestions for Authors

The author has addressed the comments from the first review and made the requested revisions. The changes are clear and complete, meeting the submission requirements.

Author Response

Reviewer 2

The author has addressed the comments from the first review and made the requested revisions. The changes are clear and complete, meeting the submission requirements.

Ans. I thank you for kindness and generousness.

Throughout this revision process, I learned a lot about autophagy and allergy.      

Round 3

Reviewer 1 Report

Comments and Suggestions for Authors

The authors have addressed my suggestions.

Author Response

Comment: The authors have addressed my suggestions.

Dear Sir 

I thank you for your kindness and outstanding perspective.

I learned a lot about autophagy and allergic inflammation. I thank you again.

Sincerely yours 

Jeoung Dooil, Ph.D.

Professor of Biochemistry

Kangwon National University 

Chuncheon 24341, Korea